# The Baculovirus Expression System Expresses Chimeric RHDV VLPs as Bivalent Vaccine Candidates for Classic RHDV (GI.1) and RHDV2 (GI.2)

**DOI:** 10.3390/vaccines13070695

**Published:** 2025-06-27

**Authors:** Yan Wang, Yiyang Fan, Ruixiang Bi, Yapeng Zhao, Wanning Gao, Derong Zhang, Jialin Bai

**Affiliations:** 1Engineering Research Center for Key Technology and Industrialization of Cell-Based Vaccine, Ministry of Education, Northwest Minzu University, Lanzhou 730030, China; wysakura616@163.com (Y.W.); 13354925037@163.com (Y.F.); gwn17679013680@163.com (W.G.); zdr_79@126.com (D.Z.); 2Key Laboratory of Bioengineering and Biotechnology of the National Ethnic Affairs Commission, Biomedical Research Center, Northwest Minzu University, Lanzhou 730030, China; 3Gansu Technological Innovation Center of Animal Cell, Biomedical Research Center, Northwest Minzu University, Lanzhou 730030, China; 4College of Life Science and Engineering, Northwest Minzu University, Lanzhou 730030, China

**Keywords:** rabbit hemorrhagic disease virus, VP60 protein, chimeric RHDV virus-like particle, vaccine, recombinant baculovirus expression system

## Abstract

Background: Rabbit hemorrhagic disease (RHD) is an acute, hemorrhagic and highly lethal infectious disease caused by rabbit hemorrhagic disease virus (RHDV), which causes huge economic losses to the rabbit breeding industry. Moreover, there is limited cross-protection between the two different serotypes of classic RHDV (GI.1) and RHDV2 (GI.2). The shortcomings of traditional inactivated vaccines have led to the development of novel subunit vaccines that can protect against both strains, and the VP60 capsid protein is the ideal antigenic protein. This study focused on developing a bivalent RHDV vaccine that can prevent infection with both GI.1 and GI.2 strains. Methodology: Baculovirus vectors containing classic RHDV and RHDV2 VP60 were co-transfected with linearized baculovirus into sf9 cells and transferred to baculovirus via homologous recombination of the VP60 gene. Infected sf9 cells were lysed, and after purification via Ni-NTA chromatography, VLPs were observed using transmission electron microscopy (TEM). In order to evaluate the immunogenicity of the chimeric RHDV VLP vaccine in rabbits, the RHDV VP60-specific antibody, IL-4, IFN-γ and neutralizing antibody titers were analyzed in serum using ELISA and HI. Results: The recombinant baculovirus system successfully expressed chimeric RHDV VLPs with a diameter of 32–40 nm. After immunization, it could produce specific antibodies, IL-4 and IFN-γ. Following the second immunization, neutralizing antibodies, determined using hemagglutination inhibition (HI) assays, were elicited. Conclusions: These data show that the chimeric RHDV VLP bivalent vaccine for immunized New Zealand rabbits can induce humoral immunity and cellular immunity in vivo, and the immunization effect of the high-dose group is similar to that of the current commercial vaccine.

## 1. Introduction 

Rabbit hemorrhagic disease (RHD), colloquially termed rabbit plague, is an acute and highly fatal infectious disease caused by the rabbit hemorrhagic disease virus (RHDV), primarily affecting rabbits [1,2]. RHDV was first identified in Europe in the 1980s, and it has had a significant economic impact on the rabbit breeding industry [3]. The classic RHDV(GI.1) is the earliest strain, which only infects adult rabbits; it was introduced into China in 1984 and rapidly spread throughout the country. With the evolution of the strain, a new genotype, RHDV2(GI.2), appeared in France in 2010, which had a wider range of infected hosts and was more infectious than classic RHDV, followed by other countries, including Australia, Italy, Germany, Spain and the USA, reporting the occurrence of the disease [4,5,6,7]. RHDV2 gradually replaced the classic RHDV as the main circulating strain, but the classic strain still exists and poses a threat to rabbit farming [8].

RHDV is a non-enveloped, single-stranded positive-sense RNA virus belonging to the genus *Lagoviruses* in the *Caliciviridae* family, with a genome length of approximately 7.4 kb [9]. It has two open reading frames; ORF1 encodes seven non-structural proteins and the nucleocapsid protein VP60 after proteolytic enzyme cleavage, and ORF2 forms the minor structural protein VP10 [10,11]. The VP60 protein is the preferred target for RHDV diagnosis and subunit vaccine development due to its surface antigenic epitope and ability to self-assemble into virus-like particles of a similar size to natural RHDV.

The traditional attenuated vaccine and inactivated vaccines can elicit host-specific immunity and stimulate the differentiation and proliferation of T cells and B cells [12,13]. However, it carries the risk of incomplete inactivation leading to large-scale outbreaks. A novel subunit vaccine is needed to prevent RHDV infection. Virus-like particles (VLPs) are hollow particles composed of one or more structural proteins of the virus that can display viral antigens and induce strong humoral and cellular immunity responses; they are also safe due to the lack of viral nucleic acids, which prevents virus replication and proliferation [14,15]. Selecting a suitable system for VLP expression is crucial for intact VLP structure and immunogenicity. The baculovirus expression system, established for decades, is advantageous for expressing exogenous proteins. Baculovirus can accommodate foreign genes of large molecular weights and allow multiple insertions, with protein post-translational modifications and folded proteins similar to mammalian cells, facilitating protein function performance [16,17]. Currently, all RHDV genetically engineered inactivated vaccines approved for production in China use the baculovirus expression system.

Previous studies have shown that even rabbits vaccinated with GI.1 can become infected with RHDV2, so we need a vaccine that also protects against them [18]. The objective of this study was to explore a faster and more efficient method for expressing the VP60 protein by co-transfecting insect cells with baculovirus vectors and linearized baculovirus. We describe the simultaneous expression of the capsid protein VP60 of both classical RHDV and RHDV2, leading to the formation of chimeric RHDV virus-like particles, which are then prepared as a bivalent RHDV vaccine to evaluate potential for immunoprotection in rabbits.

## 2. Materials and Methods

### 2.1. Cells, Viruses and Clone

Sf9 cells (Lanzhou Bailing Biotech, Lanzhou, China) were grown in serum-free insect cell culture medium (Lanzhou Bailing Biotech, Lanzhou, China). The VP60 gene of the classic RHDV (GenBank accession no. DQ205345) and RHDV2 (GenBank accession no. MN061492) were synthesized by GenScript Biological Technology Co., Ltd. (Nanjing, China) and cloned into the pQBDual plasmid. The PQBDual plasmid and qBac-III were purchased from Bacmid Biotechnology Co., Ltd. (Xianyang, China).

### 2.2. Expression and Purification Recombinant Proteins

Sf9 cells were grown in a 50 mL cell culture shaking flask and adjusted to 2.5 × 10^6^ cells per milliliter. Recombinant baculovirus was inoculated at 3 MOI, and 500 μL of the cell suspension was re-pipetted daily for observations with fluorescence microscopy. After 5 days of culture, the cells were centrifuged at 1000 rpm for 5 min, the supernatant was discarded, and the cells were washed three times with PBS. The cells were resuspended in 30 mL of lysate (500 mM NaCl, 50 mM Na_2_HPO_4_, 0.2% TritonX-100, 1% glycerol, pH 7.8), ultrasonically lysed under ice conditions for 40 min, and centrifuged at 12,000 rpm for 30 min, and the supernatant was filtered using a 0.22 μm filter (Millipore, Darmstadt, Germany). The Ni-NTA chromatography (Bestchrom, Jiaxing, China) column was equilibrated with lysate, and then, the supernatant flowed through it. Imidazole was used at 20 mM, 30 mM, 40 mM, 50 mM, 100 mM and 250 mM to wash away impurities and elute the respective target proteins, and the collected liquid was analyzed via SDS-PAGE.

### 2.3. PCR, SDS-PAGE and Western Blotting Analysis

Sf9 cells were seeded in six-well plates at 40–50% cell confluency, and the recombinant baculovirus was inoculated at 3 MOI. The viral genome was extracted from sf9 cells that were infected with recombinant baculovirus, according to the instructions of the Tiangen Virus Genome Extraction Kit (Beijing, China), and it was detected via PCR using the following forward and reverse primers shown in Table 1. Sf9 cells were lysed using insect cell protein extraction reagent (Thermo Fisher Scientific, Waltham, MA, USA). After centrifugation at 12,000 rpm for 30 min at 4 °C, both the supernatant and pellet fractions were collected. Samples were resolved via 10% SDS-PAGE and electrotransferred onto polyvinylidene fluoride (PVDF) membranes, sealed with 2.5% skimmed milk powder and incubated with His-tag antibody overnight at 4 °C. HRP-conjugated goat anti-mouse was used as the secondary antibody, and the blot was developed using picoLight enhanced chemiluminescence (Epizyme, Shanghai, China).

### 2.4. Enzyme-Linked Immunosorbent Assay

RHDV-specific antibodies in immunized rabbits were quantified using indirect ELISA. Classic RHDV-specific antibodies were detected using a commercial RHDV IgG antibody ELISA assay (LVDU Bio, Binzhou, China). The purified RHDV2 VP60 protein was diluted to 5–10 μg/mL with carbonate buffer; 100 μL per well was added to a 96-well plate, and it was coated overnight at 4 °C. Then, the liquid was discarded, and it was washed three times. The serum to be tested was diluted with PBST at 1:1000, and 100 μL was added to each well and incubated at 37 °C for 1 h; it was then washed, and goat-rabbit IgG-horseradish peroxidase (1:10,000-fold dilution) was added and incubated at 37 °C for 1 h. The plate was washed, and TMB chromogenic solution was added, at 100 μL per well, and incubated at 37 °C in the dark for 15 min, and the reaction was stopped using 1 M H_2_SO_4_ and read at 450 nm.

### 2.5. Transmission Electron Microscopy Detection

To verify whether RHDV VP60 proteins self-assemble into VLPs, the purified protein sample was subjected to an overnight dialysis process at 4 °C to remove imidazole. RHDV VLP morphology was analyzed via negative-stain transmission electron microscopy. Briefly, 10 μL of purified RHDV VLPs was applied to a carbon-coated copper grid (300 mesh) and incubated for 30 s. Excess sample was blotted with filter paper, and grids were negatively stained with 3% phosphotungstic acid for 45 s. Negative stained samples were examined via TEM.

### 2.6. Hemagglutination Inhibition Assy

Serum samples were analyzed for RHDV antibody titers via a hemagglutination inhibition (HI) assay. In the present study, we used purified chimeric RHDV virus-like particles to detect the hemagglutination inhibition titer of antibodies in serum. Here, 25 μL of serum was pipetted into a 96-well V-plate containing 25 μL of saline for 1:2 to 1:2^14^ serial dilutions, and 25 μL of 4HA chimeric RHDV virus-like particles was added to all wells except the control group and incubated at 37 °C for 20 min. The assay was performed by adding 25 μL of 1% human type O erythrocyte suspension to each well, with subsequent incubation at 37 °C for 15 min. The HI antibody titer was recorded as the highest serum dilution that prevented complete hemagglutination.

### 2.7. Immunization of Rabbits

Eight-week-old male New Zealand White rabbits were randomly divided into four groups of three rabbits each. The chimeric RHDV VLPs obtained after purification were configured into vaccines according to 4 mg low and 8 mg high doses with the addition of an aluminum hydroxide adjuvant. PBS was injected into the negative control group, whereas the positive control group received an injection of a commercial vaccine. The third and fourth groups were vaccinated with low-dose and high-dose chimeric RHDV VLPs vaccines, respectively. The first immunization was performed on 0 day, a booster vaccination was administered at 21 days after the first immunization and blood was collected before and every 7 days after immunization to isolate serum, which was used to detect RHDV-specific antibodies, HI and cytokines.

### 2.8. Cytokine Detection

Cytokine expression levels were detected using IL-4 and Interferon (IFN)-γ test kits (Enzyme-linked Biotechnology, Shanghai, China) according to the manufacturer’s instructions. The IL-4 and IFN-γ responses was assessed weekly using serum for one month. Serum was diluted, with dilution to the appropriate assay concentration, and the concentrations of IL-4 and IFN-γ in serum were determined according to the standard curve.

### 2.9. Statistics Analysis

The data analysis was performed via one-way Anova to compare the differences among the four groups using GraphPad Prism 9, and *p* < 0.05 was considered statistically significant.

## 3. Result

### 3.1. Production of Chimeric RHDV VLPs

To create a novel bivalent antigen vaccine containing both classic RHDV and RHDV2, we inserted the VP60 protein nucleotide sequence into the pQB-Daul baculovirus vector, without codon optimization, and expressed the two genotypes of the VP60 protein simultaneously using the p10 and pH promoters. Six His-tags were added to the C-terminus of the RHDV VP60 gene to facilitate subsequent purification of the protein of interest (Figure 1A). The linearized bacmid qBac-IIIG with an EGFP and the baculovirus vector containing the VP60 protein were co-transfected into sf9 cells and incubated at 27 °C for 5 days without CO_2_. The culture supernatant of sf9 cells infected with recombinant baculovirus was collected, inoculated with sf9 and infected for two consecutive generations. The transfection effect and target protein expression were observed using fluorescence microscopy. Green fluorescence in sf9 cells indicated successful infection with the recombinant baculovirus and RHDV VP60 protein expression (Figure 1B). Viral genomes were extracted from sf9 cells infected with recombinant baculovirus and uninfected cells, and VP60 gene amplification was performed via PCR after reverse transcription with classic RHDV- and RHDV2-specific primers. The results showed that there was no 1740 bp target gene band in uninfected cells, while bands appeared in infected cells (Figure 1C), confirming successful integration of the target gene into the baculovirus for RHDV VP60 protein expression. To explore the VP60 protein expression time, sf9 cells were seeded in six-well plates and infected with baculovirus at 3 MOI. Cells were collected daily for one week and lysed, and VP60 protein expression was assessed via SDS-PAGE. VP60 protein expression began 3 days post-infection and increased with an extension of the expression time (Figure 1D).

### 3.2. Souble Expression of VP60 Protein Self-Assembled into VLPs

To characterize the expression profile of the RHDV VP60 protein, supernatant and pellet fractions from lysed baculovirus-infected sf9 cells were analyzed via SDS-PAGE (10% gel) under reducing conditions. The results showed that there was a specific 60 kDa band in the supernatant, which indicated the VP60 protein found in the soluble fraction, while its presence in the pellet may have been due to incomplete cell lysis (Figure 2A). Western blotting confirmed protein expression in infected cells but not in uninfected cells (Figure 2B). After ultrasonication lysis, the supernatant was purified via Ni-NTA chromatography, and SDS-PAGE showed VP60 protein appearance at 60 kDa under 100 mM imidazole elution, with more protein with 250 mM imidazole (Figure 2C). To confirm VP60 protein self-assembly into VLPs, the purified protein was dialyzed overnight and negatively stained with phosphotungstic acid for transmission electron microscopy analysis. The results showed that VP60 protein formed a nucleic-acid-free, 33–40 nm-diameter hollow particle (Figure 2D). These results indicate successful RHDV VLP expression using the baculovirus expression system.

### 3.3. Humoral Immune Responses in Rabbits

Antibodies against the RHDV VP60 protein in the host can effectively inhibit RHDV infection. To evaluate the chimeric RHDV VLP vaccine’s ability to induce humoral immunity, we used a commercial RHDV antibody ELISA assay kit and a 96-well plate coated with the RHDV2 VP60 protein antigen to detect anti-RHDV VP60 antibodies in rabbit serum via ELISA. We purchased a commercial baculovirus vector inactivated vaccine for the positive control group. Both high-dose and low-dose RHDV VLP vaccine groups and the commercial vaccine group induced anti-RHDV VP60 antibody production, while the negative control group did not. After one week of immunization, the classic RHDV VP60-specific antibody produced in the commercial vaccine group was lower than that in the high-dose group, but the lower-dose group did not differ significantly. After booster immunization, both the chimeric RHDV VLP vaccine group and the commercial vaccine group had significantly increased classic RHDV VP60-specific antibodies, and there was no significant difference between the high-dose group and the commercial vaccine group (Figure 3A). After primary immunization, the production of RHDV2 VP60-specific antibodies was higher than that of classic RHDV VP60-specific antibodies (Figure 3A,B), but there was no significant difference between the groups during the immune cycle (Figure 3B). The HI assay showed that all immunized groups except the control group had virus-neutralizing ability one week after immunization, which reached the highest level after the second dose, with no significant difference between the high-dose group and the commercial vaccine group (Figure 3C). The results showed that the effect of the vaccine in the high-dose group was similar to that of the commercial vaccine.

### 3.4. Cellular Immune Responses in Rabbits

IFN-γ is a pleiotropic cytokine that has antiviral activity and immunomodulatory effects [19]. The results showed that in the second week after immunization, except for the low-dose group, the rest of the groups were different compared with the control, and in the third week after immunization, the low-dose group began to change. IFN-γ reached its highest level after booster immunization, and there were significant differences between the commercial vaccine group and the chimeric RHDV VLP vaccine group compared to the control group, but the difference between them was not significant (Figure 4A). IL-4 is a key regulator of humoral immunity and adaptive immunity [20]. At the third week after immunization, there were differences in all groups except for the low-dose group compared to the control. There was no significant difference in the levels of IL-4 induced between the high-dose group and the commercial vaccine group after booster immunization, but the low-dose group had significantly insufficient levels of IL-4 induced compared to the other two groups (Figure 4B).

## 4. Discussion

Since its discovery, RHDV has spread rapidly around the world and has become a major threat to the global rabbit industry [21]. Therefore, the development and improvement of vaccines to prevent RHDV infection has been carried out. However, the high variability of RHDV and its inability to culture in vitro pose challenges for vaccine development. Therefore, the development of subunit vaccines for RHDV prevention is crucial. Studies have shown that an inactivated RHDV tissue vaccine can induce effective neutralizing antibodies and prevent RHDV infection [22,23]. However, the mutation of RHDV and the weak cross-protection between various genotypes and the economic problems brought by vaccination with traditional tissue inactivated vaccines to rabbit farmers are gradually increasing, and it also reflects the limitations of traditional vaccines. Over the years, recombinant vaccines against the RHDV VP60 capsid protein have been developed using different expression systems, such as baculovirus, prokaryotic, yeast and poxvirus expression systems [24,25,26,27]. The self-assembly of VP60 into virus-like particles can elicit strong humoral and cellular immunity in vivo, which has become the focus of the development of new subunit vaccines for RHDV in the past few decades.

The Bac-to-Bac system is a commonly used platform in baculovirus expression systems, in which the modified *E. coli* contains an autonomously replicating plasmid or bacmid, and the target gene on the donor plasmid is inserted into the bacmid through transposposition, and the bacteria containing recombinant baculovirus can be obtained after blue–white screening [28]. The VP60 proteins of GI.1 and GI.2 strains were cloned into the pFastBacDaul plasmid, and the RHDV VLPs expressed by the Bac-to-Bac system had good immunogenicity and could resist RHDV infection [29]. The study found that RHDV VLPs, formed by co-infecting insect cells with two baculoviruses containing GI.1 and GI.2, were chimeras of two VP60 proteins, and chimeras have high stability [30]. In this study, the Bac-to-Bac system was not used to express the RHDV VP60 protein, but the baculovirus vector containing the VP60 protein was used to co-transfect insect cells with linearized baculovirus, and the VP60 protein was transferred into the baculovirus via homologous recombination, which reduced the screening step of positive baculovirus and greatly shortened the protein expression time compared with the Bac-to-Bac system. Moreover, the modified commercial bacmid deleted the unnecessary fragments that affect the expression of foreign genes in the bacmid, and the inserted anti-apoptotic fragments could prolong the cell life of baculovirus after infecting insect cells, and this greatly increased the expression of the VP60 protein. The results showed that the RHDV VP60 protein was correctly expressed in sf9 cells and existed as soluble protein. The chimeric RHDV VLPs formed via self-assembly were similar in shape and size to the natural virus. The chimeric RHDV VLP vaccine-immunized rabbits could produce specific antibodies against classical RHDV and RHDV2 VP60 at the same time, and the HI titer could be maintained at 1:2^4^ or higher, which stimulates the humoral immune response in vivo and activates cytokine production and initiates cellular immunity.

Although different doses of chimeric RHDV VLPs can elicit an immune response in rabbits, it is clear that high-dose chimeric RHDV VLPs are as effective as commercial vaccines, and the next research direction will be to improve the immunogenicity of RHDV VLP vaccines while exploring their stability at normal storage temperatures. VLPs can be stored at low temperatures for a long time without losing their biological function, which is a major advantage as a vaccine candidate; studies have shown that mi3 NPs, as a nanocage, can be used as a backbone protein to solve the problem of viral antigens that cannot form VLPs in vitro while improving the immunogenicity and stability of antigens, and the spontaneous formation of highly specific irreversible covalent bond coupling between SpyCatcher and SpyTag can maintain or enhance the biological activity of proteins [31,32,33]. These methods and strategies are the direction of future VLP vaccine development.

## 5. Conclusions

We explored the use of a baculovirus expression system to express classic RHDV and RHDV2 VP60 proteins simultaneously. Compared to using the Bac-to-Bac system, the screening step is omitted, and the VP60 protein expression time is shortened. The RHDV VP60 protein can self-assemble in vitro to form chimeric RHDV VLPs, and the antigenic epitopes of two different genotypes of RHDV strains can be displayed on the surface of VLPs. A bivalent vaccine prepared from RHDV chimeric VLPs can elicit an immune response in rabbits, laying a foundation for the subsequent response to the challenges brought by RHDV mutations and the development of virus-like particle vaccines. Further evaluation of the RHDV chimeric VLPs using experimental infections with both RHDV and RHDV are required to demonstrate efficacy of the vaccine prior to commercialization, regulatory approval and use in the field.

## Figures and Tables

**Figure 1 vaccines-13-00695-f001:**
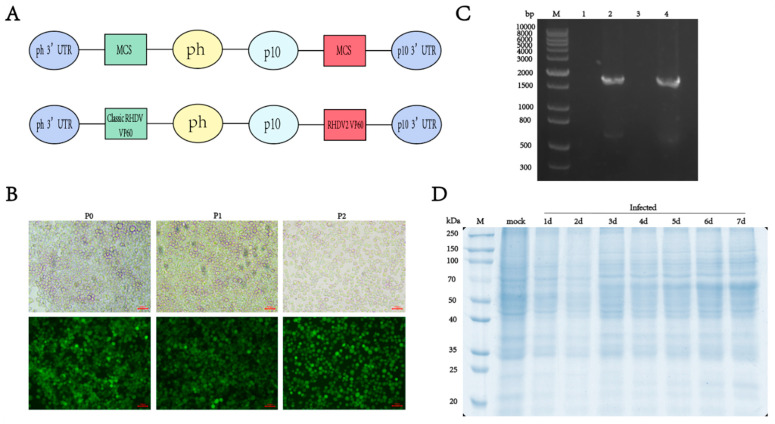
Construction and expression of recombinant VP60 protein. (**A**) Diagram indicating the structure of the recombinant VP60 protein produced for chimeric RHDV VLPs. ph: AcMNPV polyhedrin promoter, p10: p10 promoter, MCS: multiple cloning sites. Classic RHDV VP60 and RHDV2 VP60 indicate different target proteins. (**B**) Recombinant baculovirus infection of sf9 cells in P0, P1 and P2 generations. The fluorescence of EGFP was observed via fluorescence microscopy. (**C**) PCR analysis of the RHDV VP60 gene. M: DNA ladder, lane1 and lane3: sf9 cells not infected with recombinant baculovirus and amplified with classic RHDV and RHDV2 VP60 protein-specific primers, respectively, lane2: classic RHDV VP60 of gene, lane4: RHDV2 VP60 of gene. (**D**) 10% SDS-PAGE of VP60 protein expression at different times.

**Figure 2 vaccines-13-00695-f002:**
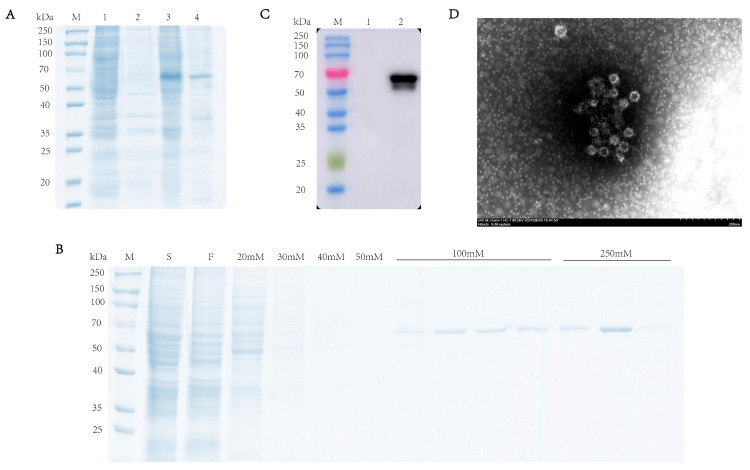
Analysis of recombinant RHDV VP60 expression. (**A**) SDS-PAGE analysis of recombinant RHDV VP60 expression. Lane1 and lane2: sf9 cells; lane3 and lane4: recombinant VP60 protein. (**B**) Coomassie Brilliant Blue-stained image of the purified VP60 protein. Lane S: cell lysate sample; lane F: fraction of flow through; lane 20 mM250 mM: fractions of wash and elution with imidazole. (**C**) Determination of VP60 protein expression. Lane1: sf9 cells; lane2: 250 mM imidazole eluate. (**D**) TEM images of chimeric RHDV VP60 VLPs. TEM analysis showed that VP60 protein assembled into VLPs with diameters of 33–40 nm. Bar = 200 nm.

**Figure 3 vaccines-13-00695-f003:**
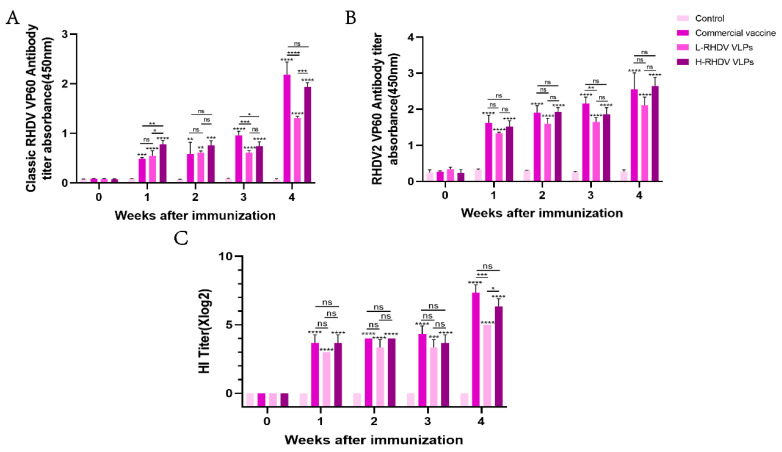
Comparative analysis of humoral immune responses induced by chimeric RHDV VLPs versus the commercial vaccine in rabbits. (**A**,**B**) Anti-VP60 antibody levels measured via indirect ELISA in serum samples of pre-immune and post-immunized rabbits. (**C**) The hemagglutination inhibition (HI) titers of serum samples from pre-immune and post-immunized rabbits were detected. ns: not significant (*p* > 0.05), * *p* < 0.05, ** *p* < 0.01, *** *p* < 0.001, **** *p* < 0.0001.

**Figure 4 vaccines-13-00695-f004:**
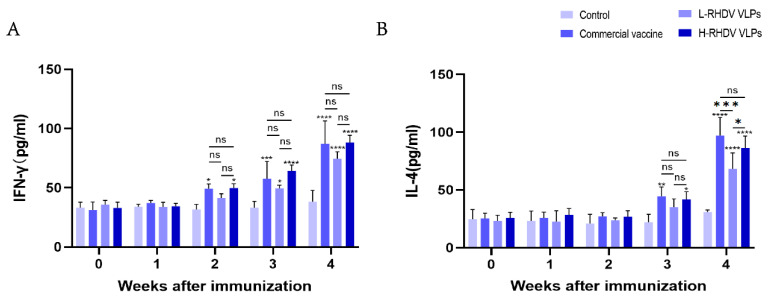
Evaluation of cellular immune responses induced by chimeric RHDV VLPs and the commercial vaccine through cytokine profiling. (**A**) IFN-γ levels in rabbit serum. (**B**) IL-4 levels in rabbit serum. ns: not significant (*p* > 0.05), * *p* < 0.05, ** *p* < 0.01, *** *p* < 0.001, **** *p* < 0.0001.

**Table 1 vaccines-13-00695-t001:** Oligonucleotide RHDV sequences of the primers used for detecting each.

Primer	Oligonucleotide Sequences
classic RHDV-F	ATGGAGGGCAAAACCCGCAC
classic RHDV-R	TCAGACATAAGAAAAGCCATTGGTT
RHDV2-F	ATGGAGGGCAAAGCCCGC
RHDV2-R	TCAGACATAAGAAAAACCATTGGTTG

## Data Availability

To obtain specific experimental data, please send an email to the first author or corresponding author.

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
