# Peer review of "The Baculovirus Expression System Expresses Chimeric RHDV VLPs as Bivalent Vaccine Candidates for Classic RHDV (GI.1) and RHDV2 (GI.2)"

_vaccines, 2025, doi:10.3390/vaccines13070695_

Round 1

Reviewer 1 Report

Comments and Suggestions for Authors

Rabbit hemorrhagic disease (RHD), a highly contagious and fatal disease of wild and domestic rabbits (Oryctolagus cuniculus), was first reported in China in 1984. Despite the development of vaccines, RHD cases are still reported. Perhaps this is due to the limited use of vaccines in some countries, as well as the genetic variability of the virus. The purposes of this work are to explore a faster and more efficient method for expressing the capsid protein VP60 of both classical RHDV and RHDV2 by co-transfecting insect cells with baculovirus vectors and linearized baculovirus. The chimeric RHDV virus-like particles caused the development of specific immune reactions after immunization of rabbits.

  1. The main note is that it has been repeatedly shown that the VP60 protein expressed using the baculovirus vector can be used as vaccines for the prevention of RHD. Due to this fact, the novelty of this study is not very clear.
  2. It is necessary to add more information (in the introduction or discussion) about previous studies on the expression of RHDV/RHDV2 VP60 using baculovirus vector, since not all of them are indicated in the references. In addition, it would be possible to add to the discussion the examples of other expression systems of this protein, their disadvantages or advantages in comparison with the baculovirus vector.
  3. Have there been recent outbreaks of RHDV in China? What genotypes and subtypes have been identified in China recently? This information should be added to the Discussion or Introduction sections if possible. Understanding the circulating strains in a country is very important for vaccine selection.
  4. What amount of protein was used to immunize the rabbits? Why was a booster immunization performed? Is this scheme planned to be used in case of commercialization of this vaccine?
  5. The main limitation of the experiment is the lack of challenge, as well as the small number of animals in the group. Although the antibody titers indicate the development of strong immunity, it would be better to infect animals to prove the effectiveness of the developed vaccine.

Author Response

Response to Reviewer Comments

  1. Summary

Thank you very much for taking the time to review this manuscript. We have carefully revised the paper based on your comments. Your valuable comments will make the article more complete.

  1. Point-by-point response to Comments and Suggestions for Authors

Comments1: The main note is that it has been repeatedly shown that the VP60 protein expressed using the baculovirus vector can be used as vaccines for the prevention of RHD. Due to this fact, the novelty of this study is not very clear.

Response1: In previous studies, the Baculovirus Expression System was used to express VP60 protein using the Bac-to-Bac system or other systems. The baculovirus expression system used in this paper uses a commercial expression system, and the modified baculovirus can successfully express a large number of VP60 proteins without codon optimization under the premise of no codon optimization, and there is no need for resistance and blue-white spot screening, and the baculovirus infection can be observed by fluorescence microscopy, which not only shortens the protein expression time but also can be observed more intuitively.

Comments2: It is necessary to add more information (in the introduction or discussion) about previous studies on the expression of RHDV/RHDV2 VP60 using baculovirus vector, since not all of them are indicated in the references. In addition, it would be possible to add to the discussion the examples of other expression systems of this protein, their disadvantages or advantages in comparison with the baculovirus vector.

Response2: Because RHDV cannot be cultured in vitro cell lines, it is not as well studied as other viruses on its own. There are only a limited number of references that can be found, but your suggestion is good, and we will try to find relevant literature to enrich the article.

Comments3: Have there been recent outbreaks of RHDV in China? What genotypes and subtypes have been identified in China recently? This information should be added to the Discussion or Introduction sections if possible. Understanding the circulating strains in a country is very important for vaccine selection.

Response3: There have been no official reports of large-scale outbreaks of RHDV in recent years, but I will check the relevant information to learn about the RHDV strains circulating in my country.

Comments4: What amount of protein was used to immunize the rabbits? Why was a booster immunization performed? Is this scheme planned to be used in case of commercialization of this vaccine?

Response4: According to the reviewed data, the vaccine is administered at doses of 4 mg and 8 mg VP60 protein. After immunizing New Zealand rabbits, we found that the effect after the first dose of immunization was not very good, and predicted that the antibody may have a downward trend, so the second dose of booster immunization was performed. The vaccine has a tendency to be commercially available in the future.

Comments5: The main limitation of the experiment is the lack of challenge, as well as the small number of animals in the group. Although the antibody titers indicate the development of strong immunity, it would be better to infect animals to prove the effectiveness of the developed vaccine.

Response5: Your suggestion is very pertinent, RHDV challenge experiment needs to be carried out in the P3 laboratory, unfortunately we are difficult to find a suitable laboratory to conduct the experiment, which is also the flaw of this article, if the follow-up conditions are available, we will consider continuing to do animal challenge protection experiments.

Finally, thank you again for all your comments on all the revisions to this article, and I wish you all the best.

Reviewer 2 Report

Comments and Suggestions for Authors

The manuscript describes the expression of chimeric RHDV2 virus-like particles (VLPs) using a Baculovirus system as potential bivalent vaccine candidates for protection against Rabbit Hemorrhagic Disease. While the study presents a topic of interest and relevance to the journal’s scope, it requires substantial revisions concerning both content and structure. The descriptions of experiments, results, and rationale are often unclear or inconsistent. Below are specific points that should be addressed, although a comprehensive revision of the entire manuscript is strongly recommended.

Abstract

  • The abstract is unclear and poorly structured. It fails to concisely present the rationale, methods, key findings, and conclusions. A complete rewrite is necessary to improve clarity and logical flow.

Materials and Methods

  • Line 105: Replace “12,000 rpm” with the corresponding relative centrifugal force (RCF, in g), to ensure reproducibility.
  • Sections 1.5, 1.6, and 1.8: These sections should focus solely on the methodology. Avoid including justifications or narrative commentary such as “we performed... to determine...”. Instead, present methods factually and succinctly.
  • Animal Use: The manuscript does not include ethical approval details or terms of use for animals, which must be provided according to standard ethical guidelines.

Results

  • Line 132: In the sentence, “We performed a hemagglutination inhibition assay to determine the RHDV HI anti-132 body titer in the serum”, clarify how many animals were used in this and other experiments.
  • Line 188: Specify the acrylamide/bis-acrylamide concentration used for SDS-PAGE.
  • Line 193: The phrase “supernatant indicated VP60 protein solubility” is misleading, as no actual solubility test was conducted. Replace all uses of “soluble” with more accurate terminology, such as “found in the soluble fraction” or “located in the supernatant.”
  • Figures 1 and 2: The gel electrophoresis and TEM images are low-resolution and blurry. Replace them with higher-quality versions that allow proper evaluation.
  • Lines 212–230: Rewrite the entire paragraph. Specifically:
    • The initial sentences claim to evaluate the vaccine’s ability to induce humoral immunity; however, no functional immune response assay was performed. The data only shows antibody recognition via a commercial ELISA.
    • Clarify the meaning of “high-dose group” and “commercial vaccine group,” including exact doses and vaccine compositions.
    • The sentence “The production of RHDV2 VP60-specific antibodies was higher than that of classic RHDV VP60-specific antibodies” is unsupported by experimental evidence. Clearly state what data substantiate this conclusion.
  • Line 237: The introductory sentence about IFN-γ’s general functions seems out of place and unrelated to the presented results. Either contextualize it appropriately or remove it.

Discussion

  • Lines 268–269: The statement regarding VP60 VLPs inducing strong humoral and cellular immunity requires a citation. Provide references to support this claim.
  • Lines 307–313: This paragraph is overly long and lacks clarity. Break it into smaller sections and clearly articulate how the findings advance scientific understanding or differ from prior works.
  • Line 324: The manuscript mentions approval for the use of human participants, yet only rabbits were involved in the study. Clarify or correct this discrepancy.

General Recommendations

  • Ensure that the manuscript complies fully with the journal’s formatting and reporting standards.
  • Improve the consistency, clarity, and accuracy of scientific terms throughout the text.
  • Verify that all claims are supported by data and referenced appropriately.
  • Conduct a thorough language and structure review, preferably involving a professional editing service if necessary.
Comments on the Quality of English Language

The writing style is clear and good, however, the authors use excerpts and language that do not express the true nature of the facts and need to be carefully reevaluated to avoid double interpretation.. 

Author Response

Response to Reviewer Comments

  1. Summary

Thank you very much for taking the time to review this manuscript. We have carefully revised the paper based on your comments. Your valuable comments will make the article more complete.

  1. Point-by-point response to Comments and Suggestions for Authors

Comments1: Line 132: In the sentence, “We performed a hemagglutination inhibition assay to determine the RHDV HI anti-132 body titer in the serum”, clarify how many animals were used in this and other experiments.

Response1: The number of animals is in the immunization schedule.

Comments2: Line 188:Specify the acrylamide/bis-acrylamide concentration used for SDS-PAGE.

Response2:Added.

Comments3: Line 193: The phrase “supernatant indicated VP60 protein solubility” is misleading, as no actual solubility test was conducted. Replace all uses of “soluble” with more accurate terminology, such as “found in the soluble fraction” or “located in the supernatant.”

Response3: Revised.

Comments4: Figures 1 and 2: The gel electrophoresis and TEM images are low-resolution and blurry. Replace them with higher-quality versions that allow proper evaluation.

Response4: I try to modify the image as much as possible to make the result clearer.

Comments5: Lines 212–230: Rewrite the entire paragraph. Specifically:

The initial sentences claim to evaluate the vaccine’s ability to induce humoral immunity; however, no functional immune response assay was performed. The data only shows antibody recognition via a commercial ELISA.

Clarify the meaning of “high-dose group” and “commercial vaccine group,” including exact doses and vaccine compositions.

The sentence “The production of RHDV2 VP60-specific antibodies was higher than that of classic RHDV VP60-specific antibodies” is unsupported by experimental evidence. Clearly state what data substantiate this conclusion.

Response5: Modified. VP60 antibody production was detected by ELISA, and neutralizing antibody levels were detected by HI assay.

Comments6: Line 237: The introductory sentence about IFN-γ’s general functions seems out of place and unrelated to the presented results. Either contextualize it appropriately or remove it.

Response6: I would like to explain why IFN-γ is tested.

Comments7: Lines 268–269: The statement regarding VP60 VLPs inducing strong humoral and cellular immunity requires a citation. Provide references to support this claim.

Response7: Cited reference 24-30 can illustrate that strong humoral and cellular immunity can be induced.

Comments8: Lines 307–313: This paragraph is overly long and lacks clarity. Break it into smaller sections and clearly articulate how the findings advance scientific understanding or differ from prior works.

Response8: Revised.

Comments9: Line 324: The manuscript mentions approval for the use of human participants, yet only rabbits were involved in the study. Clarify or correct this discrepancy.

Response9:Added.

Finally, thank you again for all your comments on all the revisions to this article, and I wish you all the best

Reviewer 3 Report

Comments and Suggestions for Authors

While the work described in this paper is not new, and VP60 from RHDV has been expressed using baculoviruses by several research teams before, it nonetheless is an advance on previous work such as that of Qi et al (2020). The main advantage being that the modified commercial bacmid deleted unnecessary fragments affecting the expression of foreign genes in the bacmid, and the inserted anti-apoptotic fragments prolonged the cell life of baculovirus greatly increasing the expression of the VP60 protein. The chimeric self-assembling virus-like particles were in sufficient concentrations to induce cytokine production, and IL-4 and IFN-γ were apparently produced after immunization and maintained at high levels

However there are several areas where the manuscript needs improvement. The first concerns aspects of the English expression and punctuation as stated in the next section. Also, there is the need to use English tenses consistently, for example using past tense to describe work done as opposed to use of present tense.

At line 149, I think it would be useful if more details on the test kits for assaying IL-4 and INF-γ could be provided (Maker, accuracy etc).

Specific changes suggested include:

Line 111, insert 'each' after detecting.

Line 140, Add 's' to make rabbit plural.

Lines 220-221. It might be better to say 'but the lower dose group did not differ significantly'

Line 256. Delete 'begin to'. The virus has spread, making the development and improvement of vaccine necessary.

Line 263. It would be better to use the word 'geneotypes' rather than 'strains'. RHDV2 is a new genotype not merely a new strain.

Figure 2D is very difficult to see properly on a computer screen. Is it possible to adjust contrast or make other adjustments to show VLPs more clearly? (e.g., remove white area of high contrast on the right-hand side of the micrograph?)

Comments on the Quality of English Language

Generally the quality of the English is fine. However, there are two problems. One is punctuation where there are no spaces following full-stops or between words and parentheses (brackets). Corrections need to be made at different places throughout the manuscript.

The second, more important issue, is that in the methods section it seems that words were used directly from maufacturers instructions for use of products rather than saying what was done using the past tense. For example, beginning at line 88, it should be stated that '500 µL was reremoved daily. Likewise, 'Ater 5 days of culture at 150 rpm, the culture was centifuged at 1000 rpm for 5 minutes and the supernatant discarded before the cells were washed three times with PBS. Changes need to be made to correct tenses to past tense consistently through the methods section of the manuscript, not just for the example given here.

Author Response

Response to Reviewer Comments

  1. Summary

Thank you very much for taking the time to review this manuscript. We have carefully revised the paper based on your comments. Your valuable comments will make the article more complete.

  1. Point-by-point response to Comments and Suggestions for Authors

Comments1:At line 149, I think it would be useful if more details on the test kits for assaying IL-4 and INF-γ could be provided (Maker, accuracy etc).

Response1: The source of the kit purchase has been added.

Comments2:Line 111, insert 'each' after detecting.

Line 140, Add 's' to make rabbit plural.

Lines 220-221. It might be better to say 'but the lower dose group did not differ significantly'

Line 256. Delete 'begin to'. The virus has spread, making the development and improvement of vaccine necessary.

Line 263. It would be better to use the word 'geneotypes' rather than 'strains'. RHDV2 is a new genotype not merely a new strain.

Response2: All revisions have been made in accordance with the comments.

Comments3:Figure 2D is very difficult to see properly on a computer screen. Is it possible to adjust contrast or make other adjustments to show VLPs more clearly? (e.g., remove white area of high contrast on the right-hand side of the micrograph?)

Response3: I try to modify the image as much as possible to make the result clearer.

  1. Response to Comments on the Quality of English Language

Thank you for your suggestion to let me know the mistakes in the writing, which I have seriously revised.

Finally, thank you again for all your comments on all the revisions to this article, and I wish you all the best

Reviewer 4 Report

Comments and Suggestions for Authors

This interesting study was about the vaccine candidate for RHD. The concept is novel, and I agree with the authors that the chimeric RHDV VLPs generated in this study could serve as the vaccine candidate.

The main concerns I have for this manuscript include:

1) The judgment on the efficacy of the animal model with challenging infections has to be determined. Without this, any conclusions are preliminary and questionable, although the protective immunity was observed in this work.

2) The second point is that, ideally, the comparison of the VLPs established on this work shall be compared with other currently used vaccines, even killed vaccines. This can be related to the animal model for the protection, and the production of Th1 and Th2 immunity.

Minor comments:

In the manuscript title, please spell out RHD;

In the Figure 4 of this manuscript, the current way to indicate statistical difference is tricky. The authors may consider using the letters a, ab, b, bc, c (different letters indicate significant difference, but between a and ab, b and bc).

Author Response

Response to Reviewer Comments

  1. Summary

Thank you very much for taking the time to review this manuscript. We have carefully revised the paper based on your comments. Your valuable comments will make the article more complete.

  1. Point-by-point response to Comments and Suggestions for Authors

Comments1: The judgment on the efficacy of the animal model with challenging infections has to be determined. Without this, any conclusions are preliminary and questionable, although the protective immunity was observed in this work.

Response1: Your suggestion is very pertinent, RHDV challenge experiment needs to be carried out in the P3 laboratory, unfortunately we are difficult to find a suitable laboratory to conduct the experiment, which is also the flaw of this article, if the follow-up conditions are available, we will consider continuing to do animal challenge protection experiments.

Comments2: The second point is that, ideally, the comparison of the VLPs established on this work shall be compared with other currently used vaccines, even killed vaccines. This can be related to the animal model for the protection, and the production of Th1 and Th2 immunity.

Response2: The commercial vaccine we use to immunize animals is a positive control for an inactivated baculovirus vector vaccine that is purchased as a positive control, but it is not known exactly how it is made due to the existence of commercial secrets.

Finally, thank you again for all your comments on all the revisions to this article, and I wish you all the best

Round 2

Reviewer 1 Report

Comments and Suggestions for Authors

The article can be published as it contains new information that may be useful for further research; however some issues need to be fixed:
Lines 19-21: Cross-protection between classic RHDV and RHDV2 is limited and not complete, but it is still observed. It is worth correcting this text.
Line 147: The amount of VP60 protein for immunization (doses of 4 mg and 8 mg) should be specified in the materials and methods.

In my opinion, booster immunization is very unprofitable for rabbit breeders. I hope that the authors will select the optimal composition of the vaccine (protein dose, possibly another adjuvant) in further studies (including the challenge).

Author Response

Comments: The article can be published as it contains new information that may be useful for further research; however some issues need to be fixed:

Lines 19-21: Cross-protection between classic RHDV and RHDV2 is limited and not complete, but it is still observed. It is worth correcting this text.

Reply: Revised.

Comments: Line 147: The amount of VP60 protein for immunization (doses of 4 mg and 8 mg) should be specified in the materials and methods.

Reply: Revised.

Comments: In my opinion, booster immunization is very unprofitable for rabbit breeders. I hope that the authors will select the optimal composition of the vaccine (protein dose, possibly another adjuvant) in further studies (including the challenge).

Reply: Thank you very much for your valuable advice, and we will continue to explore how to provide adequate protection for rabbits under the condition of one shot of immunity, such as changing adjuvants, adding some vaccine components that help stimulate immune response, etc., and strive to develop a vaccine that is more suitable for production and practical application.

Reviewer 2 Report

Comments and Suggestions for Authors

Although the authors have tried to adapt the text, it still requires specific adjustments.
1) The Abstract was not properly prepared, e.g., Results: “The results ...” This is the conclusion and not results. Please reorganize.
2) Line 107: rpm was not converted to gravity
3) Line 135-“standard protocols”-insert the reference because not all readers know the protocols and where to look. Is the explanation below the protocol?? Did you modify this protocol??
4) Item 7-Lines 145-152- The immunization process of the rabbits and the groups used remain unclear. What was administered to the four groups???
5) The modifications reported in Lines 212–230 regarding the “high-dose group” and “commercial vaccine group,” remain unclear. To ensure the scientific rigor of your study, it is crucial to quantify the dose and the vaccine composition used in these groups.

Comments on the Quality of English Language

The way English is presented when organizing sentences sequentially deserves revision.

Author Response

  1. Summary

Thank you very much for taking the time to review this manuscript. We have carefully revised the paper based on your comments. Your valuable comments will make the article more complete.

  1. Point-by-point response to Comments and Suggestions for Authors

Comments1: The Abstract was not properly prepared, e.g., Results: “The results ...” This is the conclusion and not results. Please reorganize.

Response1: Revised.

Comments2: Line 107: rpm was not converted to gravity

Response2: We appreciate the reviewer's suggestion to express centrifugation speed in ×g (RCF). However, in our experimental protocol, the speed is conventionally reported in rpm due to variations in centrifuge rotor radii across different laboratories. Since the exact rotor radius used in our study was not recorded, converting rpm to RCF could introduce inaccuracies. For reproducibility, we have retained the original rpm values.

Comments3: Line 135-“standard protocols”-insert the reference because not all readers know the protocols and where to look. Is the explanation below the protocol?? Did you modify this protocol?

Response3: There was a typographical error here, which has been revised.

Comments4: Item 7-Lines 145-152- The immunization process of the rabbits and the groups used remain unclear. What was administered to the four groups?

Response4: Revised.

Comments5: The modifications reported in Lines 212–230 regarding the “high-dose group” and “commercial vaccine group,” remain unclear. To ensure the scientific rigor of your study, it is crucial to quantify the dose and the vaccine composition used in these groups.

Response5: There is a tendency to commercialize the vaccine in the future, so the exact dosage and composition cannot be fully disclosed. Moreover, the commercial vaccines we purchased did not provide exact dosing and formulation, and they were only administered according to the dosing and immunization schedule on the instructions at the time of injection.

Finally, thank you again for all your comments on all the revisions to this article, and I wish you all the best.